# Processes, spatial patterns and impacts of the 1743 Extreme heat event in North China: from the perspective of historical documents

Le Tao[1,2], Yun Su[1,2, *], Xudong Chen[1,2], Fangyu Tian[1,2]

[1] Faculty of Geographical Science, Beijing Normal University, Beijing, 100875, China

[2] Key Laboratory of Environmental Change and Natural Disaster, Ministry of Education, Beijing Normal University, Beijing, 100875, China

*Correspondence to*: Yun Su (suyun@bnu.edu.cn)

**Abstract.** The study of historical extreme heat is helpful for understanding modern heatwaves. By collecting 63 historical documents from 3 kinds of historical materials and using text analysis methods based on keywords, grading and classification, this research recovered and analysed the processes over time, the spatial patterns of heat severity and the extreme heat impacts of extreme heat in North China in 1743. The results show the following: 1) The extreme heat of 1743 began to be noticed by people on June 22, began to kill people on July 14, and was most severe and attracted great attention from the central government between July 14 and 25. 2) Extreme heat occurred on the plains of Hebei and Shandong Provinces, and in the valleys of southwestern Shanxi. Areas on the plains east of the Taihang Mountains, such as Baoding, Shijiazhuang, and Xingtai, experienced the worst heat. These areas are also at high risk for heatwaves on the North China Plain in modern times. 3) In 1743, heat affected people, animals, plants and facilities and had the most severe impact on human deaths. The death toll in a single county reached dozens in a single day. Timely cooling and reducing exposure have been limited but necessary means to address extreme heat in both ancient and modern times.

## 1 Introduction

In the context of global warming, extreme heat and heatwave events and their impacts on human societies are receiving increased amounts of attention and research. The IPCC's Sixth Climate Assessment Report concluded that as anthropogenic global warming intensifies, there is a high probability that the intensity and frequency of extreme weather events will continue to become more extreme (IPCC, 2021). Future heatwaves are expected to be more intense, more frequent and longer lasting (Meehl and Tebaldi, 2004; IPCC, 2013; Han et al., 2022). Extreme heat, as a weather hazard, poses a significant threat to human life and health as well as socioeconomics by reducing agricultural yields, stressing health systems, decreasing labour force efficiency, and damaging infrastructure (Watts et al., 2019; Trancoso et al., 2020). Extreme heat resulted in a global

economic loss of approximately \$16 trillion from 1992 to 2013 (Callahan and Mankin, 2022). In China, the number of heat-related deaths quadrupled from 1900 to 2019 (Cai et al., 2020).

Case studies of past climate change and typical extreme events can provide insights for human society to better understand and respond to current climate change. However, there are fewer studies on extreme heat. In Europe, extreme heat in the past has been reconstructed and analysed in only a few studies or has been investigated at longer scales in conjunction with drought (Wetter et al., 2014; Orth et al., 2016; Camenisch et al., 2020). In China, reports of extreme heat are rarer than reports of other disasters such as droughts, floods and cold snaps. Research on extreme climate events in the last 2000 years in China indicates that there were 19 cases of abnormally hot summers in a large area (more than 2 to 3 provinces) of China in the past millennium (Zheng, et al., 2014a). Compared with 227 extreme droughts in North China during 137 BC-2000 AD and 76 extreme cold winters in South China during 1500-1950, there are few records of heatwaves (Zheng, et al., 2014 a). The poor reporting of heatwaves is partly because they occur on shorter time scales and are therefore more difficult to recognize in long time series (Tao et al., 2021). Droughts are usually recorded on seasonal to annual scales, whereas heatwaves have daily time scales (Deng et al., 2009). On the other hand, this may be because extreme heat is less devastating than droughts, floods and cold, which decrease food production systems, destroy homes and even overthrow dynasties (Brázdil et al., 2019; Chen et al., 2021; Han and Yang, 2021; Xu et al., 2021).

Among the few instances of historical extreme heat events, the event in 1743 during Qing Dynasty (1644-1911A.D) is typical and has a relatively large number of records in historical documents. One previous study noted that the maximum daily temperature in Beijing on the 25th of July in 1743 reached 44.4 °C (Zhang and Demaree, 2004). This value was obtained from early palace instruments and measured by missionaries in the Qing Dynasty (Zhang and Demaree, 2004). In the middle of the 18th century, western missionaries attempted to conduct meteorological observations in Chinese palaces (Udías, 1994; Domínguez-Castro, 2017). However, at that time, the measurement data were not yet fully available in Europe, and the observations in China were more discontinuous (Ren et al., 2022). The extreme heat record in 1743 only lasted for a few days, which hinders our understanding of the entire process of heatwave development. It is also important to recognize the effects of extreme heat and its impact in regions by comparison with modern cases. Moreover, extreme heat occurred during the warmest period of the Little Ice Age in the 18th century that was in a stage of rapid climate warming (Ge et al., 2013; Neukom et al., 2019). The analysis of this case can provide new insights into our current situation in the face of more frequent heatwave events.

Factual records in historical documents are good media for recovering extreme climate events in the past. The availability and effectiveness of historical documents as proxies for reconstructing past heatwave events are worth testing. China has a wide variety of historical documentary records, such as archives, local journals, and private diaries, which contain a large amount of meteorological, climatic and disaster information (Ge et al., 2005; Ge et al., 2018; Chen et al., 2020). Methods such

as regression analysis, physical modelling, grading, frequency statistics, and analogical analysis have been developed for the reconstruction of climate series based on historical documents (Zheng et al., 2014b). In reconstruction studies of extreme climate event cases, the inference of eigenvalues and spatial and temporal statistics of key elements have also been used (Hao et al., 2010; Chen et al., 2020; Chen et al., 2021). However, they have also been more often investigated in the reconstruction of droughts, storm floods, cold waves, and storm surges. There is a need to apply text-based analytical methods to the reconstruction of historical extreme heat. A more in-depth understanding of the extreme heat case in 1743 and its impacts could be achieved by detailed analyses of historical documents.

The objective of this paper is to recover the processes and impacts of extreme heat in 1743 based on records in historical documents by means of textual analysis methods. The core methodology used is textual analysis, including text-based grading and classification. The remainder of the paper is organized as follows: Section 2 describes the study area and methodology, which focuses on how to extract temporal information about heat extremes, as well as how to grade extreme heat records and classify them by impact. Section 3 reveals the results of the extreme heat development, spatial pattern of extreme heat and characteristics of heat impacts. Finally, the strengths and limitations of historical documents in recovering climate eigenvalues and societal impacts are discussed in Section 4.1. We also compare the 1743 extreme heat event with modern cases to identify heatwave-prone areas and discuss the impact of heatwaves on the population in Section 4.2.

## 2. Materials and Methods

### 2.1 Study area

According to historical records, the extreme heat in 1743 mainly occurred in North China. The main provinces involved in extreme heat were designated as the study area and included six provincial administrative units: Beijing, Tianjin, Hebei, Shanxi, Shandong and Henan (Fig. 1). The study area is approximately located between 32 and 42°N and 110 and 125°E, mainly on the lowland plain with an elevation below 1000 m, including North China Plain and Fenhe Valley. Most of Shanxi and northern Hebei have mountainous plateaus, which are below 2000 m, such as Taihang mountains on the border between Hebei and Shanxi. The study area is dominated by a monsoon climate with rain and heat at the same time. The average daily maximum temperature in Beijing in July is 30.9 °C. In 1743, the year of the case, the old channel of the Yellow River was as shown in Fig.1(b). Three types of documents were used in the study, and their location attributes are labelled in Figure 1. As mentioned in the introduction, there are very few records (from local annals) of extreme heat events in the study area during the last 500 years, and 1743 was very prominent (Fig. 1(c)). The drought and flood index sequences in the study area are also shown in comparison to provide a temporal scenario of the study area, and 1743 was a dry year in a relatively wet phase (Fig. 1(c)).

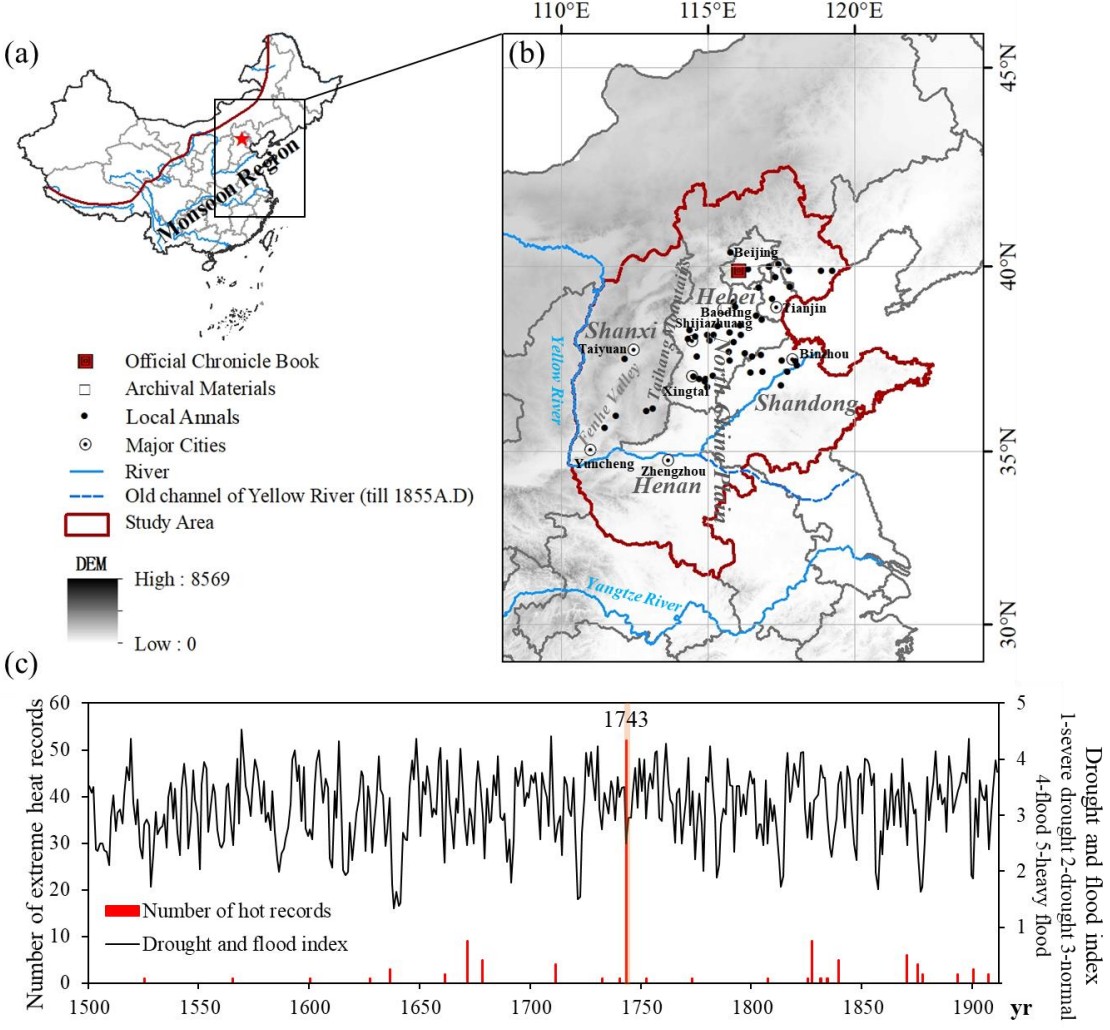

**Figure 1: Study area and data distribution in this study.**

(a) Location of the study area in China; (b) study area and data distribution; (c) temporal scenarios of the number of heat records and drought and flood indices in the study area, 1500-1911 (Tao et al., 2021; Hao et al., 2022).

## 2.2 Data source

### 2.2.1 Historical documents used

In the recovery of extreme heat information in 1743, we utilized three main types of documents. They were Local Annals, the official Chronicle book and the archival materials represented by Memos to the Emperor folded in accordion form (MEMOs for short). Three documents covered local and centralized perspectives, each of which has its own advantages.

(1) **R1**. Local annals have location information and can present the spatial distribution of extreme heat. They also recorded the feelings and impacts of humans, reflecting the severity of heat. The records of local annals were first extracted from the A Collection of Meteorological Records of the Last 3,000 years in China (Zhang 2004). This is a classic compendium for the study of China's historical climate. It excerpts 7,930 materials from local records, anthologies, and other sources. Meteorological records from the 13th century B.C. to 1911 A.D. were chronologically compiled. The records related to the

heat event of 1743 were mainly derived from various local annals. Then, traceability verification of each record was performed by referring to the digital local records collected by the National Library of China[1].

    (2) **R2**. The Factual Record of the Qing Dynasty (Qing Shilu) is the official compiled chronicle book from the Qing Dynasty. It recorded the emperor's daily dealings and measures released verbally, in which heat conditions and measures to address them in 1743 can be found. The volume of the Qing Shilu is large, but scholars have organized records related to climate, disasters and impacts into a book. The book is named A Compilation of Climate Impact Data from the Factual Record of the Qing Dynasty and was published by the Institute of Natural Resources and Environment of the Chinese Academy of Sciences (CAS,2016). The heat-related records of 1743 in this study are from this book.

    (3) **R3**. MEMOs were letters from local officials to the emperor to periodically report local conditions, including weather and disasters. They were widely used in the Qing Dynasty. Today, most of them are housed in the First Archives of Chinese History, and some are scattered in the National Library of China, Peking University, and the National Palace Museum in Taipei. The MEMOs used in our research are copied from the First Historical Archives of China[2].

    The records of extreme heat were extracted via keywords. The records of extreme heat in historical documents are mainly about human perception, such as 热 (hot), 暑 (hot) and 熏灼 (scorching). In addition, there are many records of people dying from heat stroke, and they are usually recorded as 暍死 (die of heatstroke). Using these keywords of heat and human death from heat stroke mentioned above, extreme heat records from historical documents were extracted. Notably, because summer drought occurred at the same time in 1743 (Xiao et al., 2012), heat records are often accompanied by drought. However, separate drought records were not considered as long as they did not indicate heat or that people died of heatstroke.

    Finally, a total of 51 records of extreme heat were obtained from R1, covering five provincial-level administrative regions. Among them, 1 was in Beijing, 4 were in Tianjin, 33 were in Hebei, 8 were in Shandong and 5 were in Shanxi. In addition to recording the sensation of heat and of people dying from heat stroke, R1 also recorded other impacts of heat. Ten and two extreme heat records were obtained from R2 and R3, respectively (Table 1). The records in R3 were obtained from the report of the Zhili governor, whose residence was in Baoding, Hebei Province (Fig. 1). Since R3 was staged submitted, there were fewer records on heat.

**Table 1: Historical documents used in the recovery of the 1743 extreme heat**

| | Source Types | Source | Site | Count | Examples |
|---|---|---|---|---|---|
| **R1** | **Local Annals** | Annals of counties, prefectures and states. | 51 counties or cities | 51 | *In the fifth lunar month, there is severe drought and **bitterly heat**. The soil and stone were scorched, and the metal on the roof melted. Many people died of heat. ——Local annal of Tianjin* (Supplement 1-R1-2)<br>*In the fifth lunar month, the **hot wind seemed to be baking**. **Many people died of heat**. The wheat was withered and the autumn grain could not be sown. ——Local annal of Xianxian* (Supplement 1-R1-27) |

---

[1] *http://read.nlc.cn/allSearch/searchList?searchType=12&showType=1&pageNo=1*

[2] No. 9, Qinian Street, Dongcheng District, Beijing, China (*https://fhac.com.cn/index.html*)

| R2 | **Official Chronicle book** | *Factual Record of Qing Dynasty (Qing Shilu)* | Beijing | 10 | *The emperor said it is extremely hot lately in Beijing and he was afraid that there would be many people getting heat stroke……—— Qing Shilu* (Supplement 1-R2-4) |
| R3 | **Archival materials** | Memos to the Emperor | Baoding, Hebei | 2 | *Since the half month ago,* **it had been scorching** *and the drought is worrying. There were* **many people suffering from heatstroke on the road***, and the people were very afraid of death. ——MEMO from Shen Qiyuan in Baoding* (Supplement 1-R3-2) |

All the original textual material related to the 1743 extreme heat event was digitized and organized in Supplement 1. The records in R1 have spatial attributes that allow for the spatial presentation of information. The records in R2 and R3 have more specific time information because of their recording resolution down to the daily level, which can help obtain some details of the process of the heatwave.

**2.2.2 Other materials**

To determine the precipitation conditions during the high-temperature period, Yu-Fen-Cun[3] in June and July of 1743 in the study area were also copied from the MEMOs of the First Historical Archives of China. Yu-Fen-Cun documented the rain infiltration depth after a rainfall event (Ge et al., 2005). We utilized this material to determine when large-scale precipitation occurred. The specific process is described in Supplement 2. The early daily instrumental measurements in a study by Zhang and Demaree (2004) are also referenced by us.

**2.3 Methods**

Textual analysis was the main tool of the study. To restore a more complete picture of the extreme heat of 1743, we mainly needed the following information from the text. They were temporal (date) information; information on the severity of the heat; and categories of heat impact.

**2.3.1 Determination of dates**

Historically, official and folk dates were recorded in various forms in China. Converting the dates in the raw texts in a uniform manner in a modern calendar is an important task when processing Chinese historical documents. Among them, the most common and easiest to convert is the lunar calendar. In addition, there are also dates recorded according to festivals or solar terms. Examples of data conversion are shown below:

a) For those recorded in the lunar calendar, the Chinese Calendar for Two Thousand Years: 1-2060 (abbreviated as Rf1) was utilized for direct conversion to Gregorian calendar dates. For example, the local annal of Rongcheng stated, "Due to

---

[3]Yu-Xue-Fen-Cun is a kind of agricultural weather record in Qing dynasty archives, which recorded the depth of infiltration after each precipitation (Yu-Fen-Cun) and the snow depth (Xue-Fen-Cun). It can be obtained from the First Historical Archives of China. (See detailed the introduction in Ge, et al. 2005).

extreme heat and drought, many people died of heat stroke from the 24th day in the 5th lunar month to the 5th in the 6th lunar month." (Supplement 1-R1-21). By consulting Rf1, it is clear that the 24th day in the 5th lunar month was July 15, and the 5th day in the 6th lunar month was July 25.

b) When date is recorded in terms of the solar terms or festival, it is determined according to general knowledge. For example, the MEMO from Zhili Governor Gao Bin reported, "Since the first solar term in the 6th lunar month, I feel very hot." (Supplement 1-R3-2). In Chinese folklore, the first solar term in the 6th lunar month refers to Minor Heat on approximately July 8.

c) Some were described as specific phases in a given month. Their dates were determined based on text details.

We focused on the following three types of dates. First, the dates of the beginning or end of heat in the records indicate when heat was perceived and remembered. The second are the dates of heat-related deaths, as this is a landmark event that reflects heat became more severe. In addition, the records themselves have date attributes, such as the time of signing of the MEMOS and the date when the emperor verbally released measures, which, together with the information in the content, provide information about heat. At the same time, the act of the emperor verbally enacting measures against extreme heat also indicated that the heatwave had become more severe.

**2.3.2 Grading the records by heat severity**

Although there was a total of 51records from R1, i.e., 51 locations where heatwaves were recorded, the differences in their textual descriptions of the events indicate differences in severity. To identify whether this difference represents a spatial inhomogeneity of extreme heat, we categorized the heatwave descriptions at the 51 sites into different degree classes according to certain principles. When grading by textual description, we considered two main aspects.

(1) **Whether heat-related deaths occurred**. The heat that kills people is considered to be more severe than the heat that does not kill people, and more deaths indicate a more extreme heat.

(2) **Use of words (semantic difference)**. First, the adverbs of degree in records made sense and could not be ignored. For example, descriptions such as "毒热 **(toxically hot)**", "熏灼 **(scorching)**" and "苦热 **(bitterly hot)**" are believed to be more extreme than those of ordinary "热 **(hot)**" and "甚暑 **(very hot)**". Second, rich sentences with vivid metaphors and more text may indicate more severe heat. For example, the local Annal of Anguo stated, "五月大热，屋壁地榻什物尽如火炙，人多热死，连六七日 (It was particularly hot in the fifth lunar month. The walls, floors, beds and all kinds of furnishings were as hot as fire. Many people died from the heat. This lasted for six to seven days)" (Supplement 1-R1-23). This record is very detailed and vivid, showing that the heat was even more extreme and impressive.

The 1743 heat records from R1 were divided into 4 levels (Table 2). Among them, Grade I indicates that people perceived unusual heat. Level II indicates that heat-related deaths occurred. In Level III, heat-related deaths were common, and in Level

4, heat was more intense and more impressive than in Level 3. The number of records for each of the four levels was 4, 7, 22 and 18, which is somewhat consistent with a skewed distribution of extreme events.

**Table 2: Criteria for grading heat event records from local annals (R1)**

| | Phenomenon | Example | Amount |
|---|---|---|---|
| I | There are just normal and clear descriptions about heat. | *In summer, it was very hot. ——Local annal of Quwo (Supplement 1-R1-42)* | 4 |
| II | There are descriptions of heat and records of population deaths. | *It was great hot in the last ten days of the 5th lunar month, there were people that died from heat. ——Local annal of Quwo (Supplement 1-R1-46)* | 7 |
| III | There are descriptions of heat and records of large population deaths, but the records are relatively brief. | *In the sixth lunar month, there was drought and scorching heat, and many people died from heat. ——Local annal of Quzhou (Supplement 1-R1-34)* | 22 |
| IV | There are vivid descriptions about how hot it was and there were large numbers of population deaths. The adjectives to describe heat are rich and descriptions about deaths show that populations death were very common. | *It was particularly hot in the fifth lunar month. The walls, floors, beds and all kinds of furnishings were as hot as fire. Many people died from the heat. This lasted for six to seven days. ——Local annal of Anguo (Supplement 1-R1-23)* | 18 |

### 2.3.3 Classification of impact records

A total of 51 records from R1 documented the impacts of extreme heat and its timescale. Among these, human heat stroke
and death were the most common. We categorized the impacts of heat. Depending on the affected objects, the impacts of the 1743 extreme heat event can be divided into four categories: (1) human, who experienced heat stroke or died; (2) animal, which died or was physically impaired; (3) plant, which dried out or died from heat; and (4) facility, which was damaged by melting. In Category (2), when crops were affected, they in turn affected harvests and food prices, leading to hunger. It should be emphasized that only if the text clearly indicated that the crop was damaged by heat was it counted as Category (2).
However, crop failures are more likely to be caused by droughts and hot winds. It is difficult to identify how much was influenced by heat. We also discerned the time scales over which different impacts persisted. We then counted the number of impact records that still existed for each month on a monthly basis. For example, impact records of population deaths, damage to facilities and animal injuries were counted in the month in which they were recorded; impact records of crop failures were counted in the month in which they were harvested; and economic and social impacts on this basis were counted in the month
in which they were recorded. Table 3 shows examples of records and the number of records for each category of impact.

**Table 3: Classification of the impacts of the 1743 heatwave and examples in local annals**

| | 1st order impact (Number of records) | 2nd order impact (Number of records) | Examples | Timescale of impact |
|---|---|---|---|---|
| 1 | **Human(47)** | | *In the 6th lunar month, no rain, very hot, **many people died of heat.*** *In early part of the 6th lunar month, the heat and drought were very severe, and the air was hot as if woods were burning. In the early days of Sanfu, **people died immediately after heat stroke.*** | In 1 month |
| 2 | **Animal(5)** | - | *It's very hot in the 6th lunar month, many people and **livestock died of heatstroke***. *From spring to the 6th lunar month, there was no rain, very hot summer, **chickens didn't stay in their nests to incubate.*** | In 1 month |
| 3 | **Plant(10)** | Grain Harvest(7) Grain Price(3) Starvation(3) | *Great drought. In the 5-6th lunar month, hot wind was inflammatory as if burning. Many people died of heat. The **wheat was all withered** and the **autumn grain could not be sown**. The emperor issued an edict to feed **the hungry** form the **8th lunar month to the 5th lunar month of next year**.* *A thousand miles of drought. Indoor furnishings were hot. The wind was hot like burning, and many of the **trees towards the southwest were dead. The harvests of both early and late rice were 30%, and lacked flavour. Part of rice grains are black.** The grains of sorghum and yellow rice were not full and the hot wind destroyed the bean seedlings. **The price of grain went up to 150 wen per dou***. In the 6th* | to the 5th lunar month of next year |

| | | | | | |
|---|---|---|---|---|---|
| 4 | Facility(2) | - | *lunar month, **many people fled from South of Tianjin and Wuding**. Many passers-by died of heat. There was contamination in the well and the water was too shallow for the boat to run.* *In the fifth lunar month, there was severe drought and bitter heat. The soil and stone were scorched, **and the metal on the roof melted**. Many people died of heat.* | In 1 month |

\***Wen** is a unit of currency in the Qing Dynasty. **Dou** is a unit of weight. One **dou** is equivalent to 6.25 kilograms.

## 3. Results

### 3.1 Processes over time of the 1743 extreme heat event

The calendar format was chosen to show the progression of extreme heat events. The data obtained according to Section 2.3.1 are presented on the calendar (Fig. 2). There were 9 records in R1, 4 in R2 and 2 in R3 indicating more specific periods of extreme heat according to the recording date, solar term, or phases in the month. Almost all of the records with specific end dates indicated that the heat lasted until July 25, except for one R1 record that was particularly hot on July 30. For the beginning of the extreme heat in R1, one record clearly indicated that it started on July 15, 2 records indicated that it started on July 17,

1 record indicated that it started on July 19, 2 records indicated that the heat began at the beginning of Sanfu[4] (July 19 of the year), and another record indicated that the heat began at the end of the 5th lunar month (July 20 is the last day of the 5th lunar month). For the records from R2, 1 indicated that heat began at the end of the 5th lunar month. Another record even indicated that the weather had been significantly hotter than in previous years since the summer solstice (June 22). For the records from R3, the situation of passers-by and farmers dying of heat continued for at least a few days before July 21, and since Minor

Heat, July 8, the weather had been extremely hot and hard to withstand. More detailed text about the timing of the extreme heat can be found in the Table in Supplement 2.

---

[4] Part of the Chinese folk calendar. It refers to the hottest time of the year. It is divided into three periods with 30 or 40 days in total, with 10 days in the first period, 10 or 20 days in the second period and 10 days in the third period.

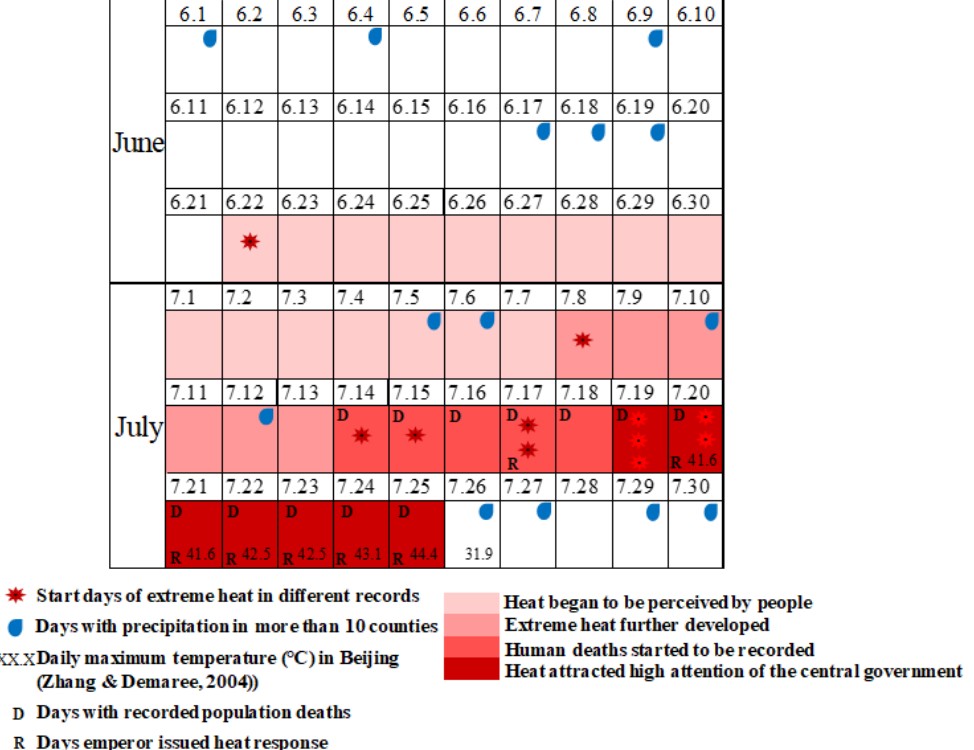

**Figure 2: Processes over time during the 1743 extreme heat event (in the form of a calendar).**

There were clear records of population deaths beginning on July 14 or July 15 and continuing through at least the end of the heatwave on July 25. R2 shows that on the 26th day of the fifth lunar month (July 17), Emperor Qianlong verbally deployed countermeasures in regard to the hot weather for the first time. From the 29th day of the 5th lunar month to the 5th day of the 6th lunar month (July 20 to July 25), the emperor worried about hot weather almost every day. Yu-Fen-Cun also indicated almost no precipitation from July 12th to July 26th.

Taking the above information into consideration, we divided the development of extreme heat in 1743 into four stages, indicated by different shades of red (Fig. 2). June 22 was the earliest occurrence of heat in the record, after which the weather may have been on the hotter side. On July 8, the second record indicated that significant heat appeared, and it is possible that the precipitation of the previous two days did not curb the momentum of the hotter weather. From July 14 onwards, heat-related deaths continued to be recorded. On July 19 and after, deaths continued, heat records increased significantly, and extreme heat began to be taken seriously by the central government.

**3.2 Spatial pattern of heat severity in 1743**

The map below shows the spatial distribution of extreme heat records and their different severity levels in R1 (Fig. 3). Extreme heat in 1743 occurred mainly on the plains of Hebei and Shandong Provinces, and in the valleys of southwestern Shanxi. Locations with more severe impacts and abundant records were mainly located on the premountain plains east of the Taihang Mountains, e.g., major cities in the eastern foothills of the Taihang Mountains, such as Beijing, Baoding, Shijiazhuang,

and Xingtai, all recorded more severe heatwave events (Level IV). This may have been the consequence of a combination of weather and topographic factors. Level I and II records with relatively mild impacts and abbreviated records were mainly found on the fringes of the records but were also distributed in the core area. The record classes in the text reflect to some extent the distribution of the degree of heat but were also influenced by anthropogenic factors.

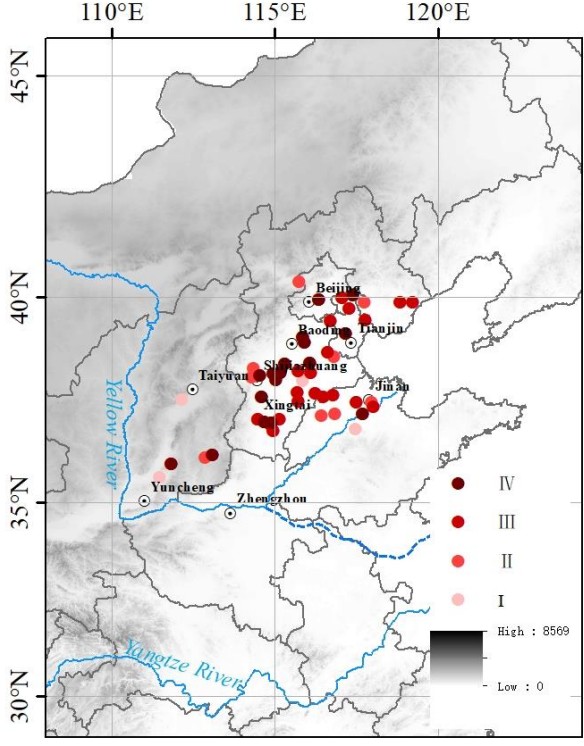

**Figure 3: Spatial pattern of heat severity.**

### 3.3 Impact of the 1743 extreme heat event

The spatial distribution of the impact on humans is generally consistent with that of extreme heat. Of the 51 records from local annals, 47 involved human deaths. The records of animal and facility impacts, although limited to 5 and 2, respectively, were scattered throughout the study area (Fig. 4(b), Fig. 4(d)). This suggests that they may not have been isolated phenomena

in 1743 but were omitted from the record because they were not emphasized. Ten records on plant impacts were distributed in the eastern part of Hebei Province and the northern part of Shandong Province, which appeared to be the result of compounding with drought or dry hot wind. Some records indicated that extreme heat caused crops to wither or soil moisture to deteriorate, resulting in crop failure or inability to sow. These factors then affected regional food prices and resulted in famine and the migration of hungry people.

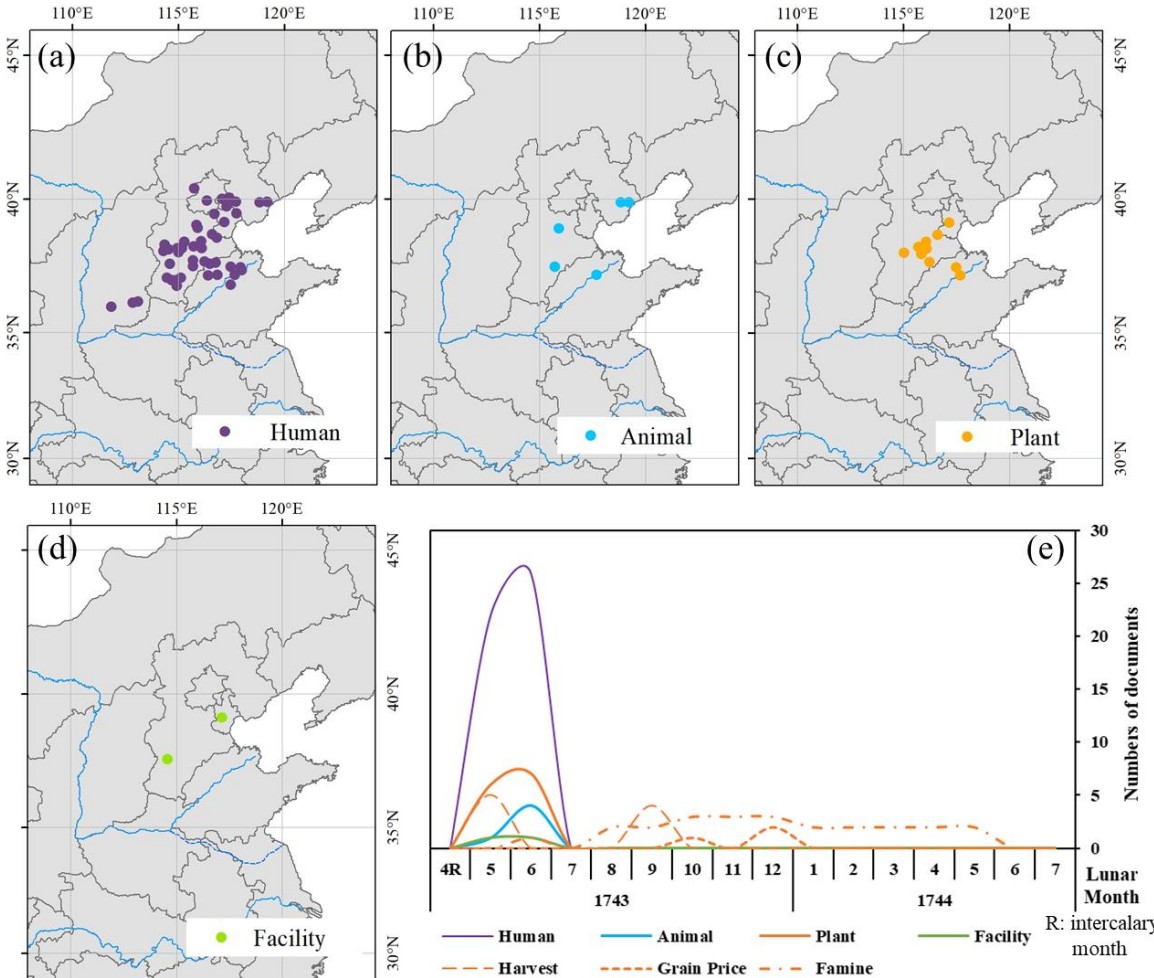

**Figure 4: Spatial distribution and time series of extreme heat impacts records in 1743.**

classified by impact object (see Tab 3): (a) Human; (b) animal; (c) plant; (d) facility; and (e) time series of impacts.

The subsequent effects of crop failure are also documented in the extreme heat event in 1743. They led to higher grain prices, famine-related migration and other social impacts. There were 7 records from local annals documenting grain harvest failure due to hot weather. Five records reported reduced harvests of wheat (usually harvested in the 5th lunar month), and 4 reported reduced harvests of autumn grain (usually harvested in the 9th lunar month). Of the three records of increasing grain prices, two recorded an increase in annual grain prices, and one recorded an increase in grain prices due to autumn crop failure. There were 4 records in total mentioning people who starved. Among them, 2 indicated that relief measures to appease the starving lasted from the 8th lunar month to the 5th lunar month of the next year, 1 record indicated that starving people were exempted from taxes that year, and 1 record directly stated that during the 6th–7th lunar month, many starving people fled to other places. Together, the above records suggested that the food security impacts of the 1743 heat event lasted for more than a year in the social system, which was longer than that of the massive outbreaks of human heat deaths.

## 4. Discussion

### 4.1 What else can be learned from the textual records in the 1743 extreme heat event?

In addition to the above information, which can be obtained more directly from the literature, we can further extract other information related to extreme heat from historical records through inference and generalization. However, confined to the characteristics of written records, we found that some influence-related patterns can be found, but quantitative extrapolation of climate elements may be difficult.

### 4.1.1 Air temperature inference in the 1743 extreme heat event

It is difficult to quantitatively reconstruct meteorological elements using qualitative historical documents. Quantitative inferences can be made by extrapolating from the present to the past, but there is uncertainty.

In the reconstruction of extreme heat, we also attempted to infer the temperature range from the human body's heat perception and heat-stroke response. There are several indices and models used to measure human perception and health risks under different environmental conditions in different countries or regions. We utilized NOAA's heat index and its danger

warning classifications under different air temperatures and relative humidities (Table 4). Based on the descriptions of massive heat-related deaths, it is clear that the Level III and IV counties (in Section 3.2) reached the warning level of "Extremely Hot". Using 40-50% humidity, which is the multiyear average relative humidity level for July in the study area, a range of possible temperatures was estimated. The calculations can be performed directly on the NOAA website[5]. The results indicated that a total of 40 counties with Levels III and IV were likely to have temperatures of 40 °C or higher in the 1743 extreme heat event.


**Table 4: Classifications of danger warnings of the heat index (NOAA)[6]**

| Classification | General effect on people | Estimated lower limit of temperature* |
|---|---|---|
| Very Warm | Fatigue POSSIBLE with prolonged exposure and/or physical activity. | |
| Hot | Sunstroke, heat cramps, or heat exhaustion POSSIBLE with prolonged exposure and/or physical activity. | 30~32 °C |
| Very Hot | Sunstroke, heat cramps, or heat exhaustion LIKELY, and heatstroke POSSIBLE with prolonged exposure and/or physical activity. | 36~37 °C |
| Extremely Hot | Heat/Sunstroke HIGHLY LIKELY with continued exposure. | 40~43 °C |

*Calculated at 40~50% humidity.

Although the estimated temperature agrees well with Zhang's results (Zhang, 2004), unlike physical, chemical or biological proxies, textual records cannot help establish a quantitative equation with meteorological elements. Danger warning levels are artificially assigned classifications of general effects on people who do not correspond in a physical sense to

individual responses.

---

[5] *https://www.wpc.ncep.noaa.gov/html/heatindex.shtml*

[6] Heat Index Chart (*https://www.noaa.gov/sites/default/files/2022-05/heatindex_chart_rh.pdf*)

### 4.1.2 Impact transmission chains in the 1743 extreme heat event

Although the subjective nature of written documents limits their potential for quantitative reconstruction, subjectivity also means that they focus on human experience and the impact of extreme weather and climate on human society. Thus, more details about human societies, such as how extreme heat transmission impacts, can be gleaned from historical documents.

Based on the records obtained in our research, it is possible to see the aspects affected by the extreme heat of 1743 and how they were transmitted and retained in human society. The results of Section 3.3 show that the main impact transmission chains were as follows: 1) heat→human death; 2) heat→animal physiological disorder or death; 3) heat (combined with drought or dry and hot wind) → crop damage → reduced food production → Hungry people/raising food prices; and 4) heat → facility damage. Impact Chain 3) was special because it went beyond the physical effects of heat on living things or objects. The effects were magnified in society. It has been noted that the impact transmission chain of food security is the main way that extreme climate events have historically affected society (Fang et al., 2015). This is because food production was a core of the livelihood of people in ancient China. Its impact could further influence higher levels of society. Actually, because of the extra attention that it is given, it occupies a large part of the record. The preference for recording socioeconomic impacts is consistent with the ancient Chinese concept of agriculture-based development.

### 4.2 Comparison with modern cases

Reconstructing past cases and comparing them with modern situations can provide more insights into extreme heat. Based on the results of this study, comparisons can be made in two directions. One is through spatial comparisons to determine whether those areas that were hotter in 1743 behaved the same way in modern times, and the second is to explore responses to the effects of high-temperature heatwaves by comparing the main effects of heatwaves in terms of the number of population deaths.

### 4.2.1 Areas with more severe heat risk

With the increasing frequency of extreme heatwaves, areas suffering more severe heat in different given cases will be at greater risk in the future. According to the results of Section 3.2, severe heat was recorded in cities along the Taihang Mountains, such as Beijing, Tianjin, Baoding, and Shijiazhuang. It seems that in recent years, these cities have also often been reported simultaneously in extreme heat events in North China.

To explore whether the extreme heat in North China in 1743 was spatially consistent with that in modern times, three typical years, 2000, 2002 and 2022, were selected for comparison according to the results in Section 3.2. The daily maximum temperatures in summer (from June to August) at 156 meteorological stations in the study area in 2000, 2002 and 2022 were obtained from the China Meteorological Data Center. We first defined the periods of extreme heat in 2000, 2000 and 2002. A heat event began when one or more stations recorded a maximum daily temperature higher than 38 °C and ended when no

stations recorded a maximum daily temperature higher than 38 °C in the following 5 days. The results showed that in 2000, the heat period was long, from the end of June to July 26, and in 2002, it lasted from July 10 to July 19. In 2022, the heat event was the earliest, occurring mainly in mid to late June (Fig. 5). Fig. 5 shows the locations of the Level IV records (in 1743) where Tmax[7] exceeded 40 °C (in 2000, 2002 and 2022) during the period of extreme heat. Although extreme heat occurred at different times, the spatial characteristics of the Tmax distribution were similar. The black boxes on the map indicate areas where severe heat occurred in all four years, and they were mainly on the plains east of the Taihang Mountains. This region is a densely populated and economically distributed area of the North China Plain, and the risk of urban heatwaves in these places is of concern for the future.

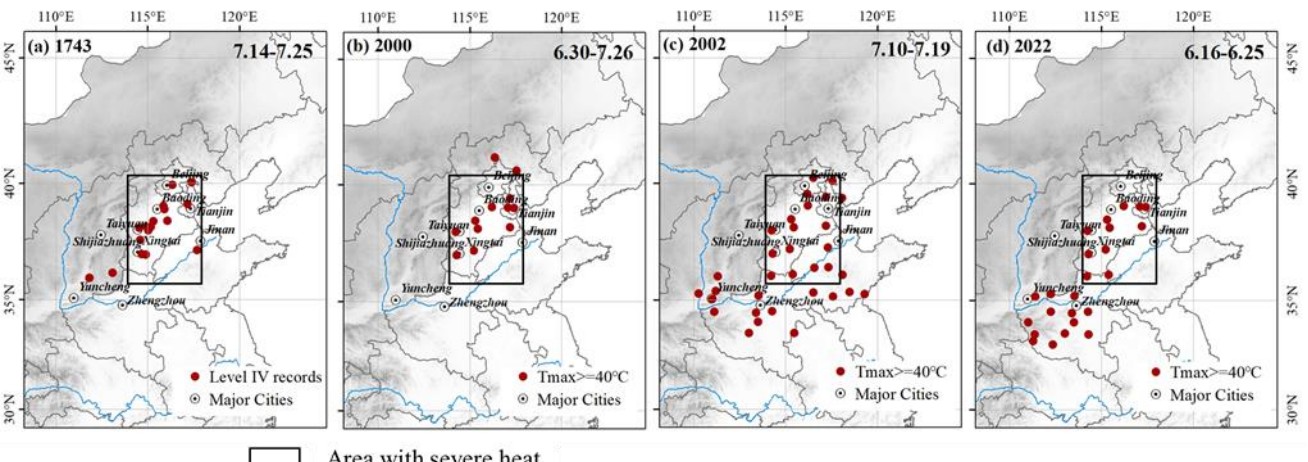

**Figure 5: Spatial distribution and time series of extreme heat impacts in 1743.**

### 4.2.2 Population deaths due to extreme heat

Humans are the main victims of heatwave disasters. Population health and life safety are core concerns in extreme heat disasters. Of all the effects in the extreme heat of 1743, human deaths were undoubtedly the most conspicuous and shocking. Descriptions of the people who died, such as "People died suddenly from heat stroke" (Supplement 1-R1-11&12) and "when it was extremely hot, people died instantly once they touched the burning air" (Supplement 1-R1-6), show that people suffered severe heat sickness in 1743 under high temperature. A letter from a French missionary, A. Gaubil, stated that between July 14 and July 25, 11,400 people had died of heat in and around Beijing (Zhang and Demaree, 2004). The spatial range in this investigation was undefined, but 2 records from local annals described the number of deaths as "dozens of people per day" (Supplement 1-R1-11 and 12). This indicates that the number of heat-related deaths in a single county was significant. Compared to modern extreme heat cases, the scale of heat deaths in historical times was considerable (Table 5).

**Table 5. A comparison of the mortality of the 1743 heat event with that of typical modern heatwaves**

| Year | Region | Days when daily maximum temperature was above 35 °C | Number of heat-related deaths |
|------|--------|------|------|
| 1743 | North China | At least ~ 12 days: 7.14-7.25 | 11,400 in Beijing and surroundings |

---

[7] Tmax refers to the maximum value of the daily maximum temperature in the current month.

| 2003 | France | 15 days: 8.4-8.18 | 3,548 (Alain et al., 2006) |
| 2013 | Shanghai, China | 33 days: 7.2-7.4/7.8-7.11/7.12-7.31/8.3-8.12 | 167 (Sun et al., 2014) |
| 2017 | England | 10 days: 6.17-6.23/7.5-7.7 | 1,489 (Rustemeyer and Howells, 2021) |

During the Qing Dynasty, people in North China mainly engaged in agriculture, which means long hours of outdoor work even in summer. Long-term exposure to high temperatures was the main cause of death. Even indoors, the threat of high temperatures is high due to lack of cooling facilities. In the face of extreme heat, individuals and governments have very limited means of coping with. Timely cooling and reducing the exposure of vulnerable populations are feasible approaches. There was a practice of storing ice in winter and using it in summer during the Qing Dynasty. However, ice was a luxury item and available only in a few places for the elite. The records of "giving ice soup and medicine" and "setting up ice factories and distributing medicine for relief" only appeared in Beijing and Tianjin, and the rest of the county town did not seem to have such conditions. Much-needed cooling measures after heat stroke were difficult to achieve. Measures such as "build pergolas" and "ordering work to stop", recorded in the Factual Record of the Qing Dynasty, were used to reduce the exposure of special groups.

Although the number of deaths has decreased compared to that in ancient times, the threat of heat to human health is still serious in modern times. Studies have shown that the risk of death in a population increases rapidly and nonlinearly at high temperatures (Gasparrini et al., 2015). The excess mortality of people with cardiovascular and respiratory diseases due to high temperature is very significant (Cheng et al., 2019). Once a person experiences heat apoplexy, the risk of death is very high. In 2022, many parts of the Northern Hemisphere experienced extreme heat. In the absence of air conditioning, many Europeans sought shade, fountains or pools to cool provisionally. Most of the time, people are helpless in the face of extreme heat. Reducing going out and paying attention to people who work outdoors and those who have underlying diseases are limited but necessary measures for addressing extreme heat.

**4.3 How the rest of the Northern Hemisphere behaved during the summer of 1743**

In this section, we discuss whether the extreme heat of 1743 was remotely related to that of other regions by comparing proxy data from multiple regions and whether these regions shared similar climatic mechanisms or contexts with modern heat events. Modern studies on heatwave mechanisms have revealed that extreme heat events in northern China and North America or Europe are often synchronized. The reason behind this is that westerly wind-belt fluctuations in the Northern Hemisphere generate multiple cut-off low and high pressures, and high-pressure ridges often form on continents, thus causing synchronized high temperatures on different continents. For example, in the summer of 2022, very intense and extreme heat occurred simultaneously in northern China and Europe. These studies have also indicated that global warming has caused the westerly wind belt to move northwards and become narrower, and it is more likely that multiple high-pressure ridges form across

360    continents, which may be the cause of the current high temperature extremes in the Northern Hemisphere (Deng et al., 2018; Kornhuber et al., 2019; Kornhuber et al., 2020; Yang et al., 2021).

    Although higher-resolution heatwave records are relatively difficult to obtain, it is possible to capture information of the 1743 summer event from a long time series. We collected 17 paleoclimatic reconstruction series from the Northern Hemisphere from NOAA (https://www.ncei.noaa.gov/access/paleo-search). There were 5 series from Asia, 6 from Europe and 6 from North

365    America. The reconstructed indices of these studies were the temperatures of partial months of the warm season (Mar-Sep), summer (Jun-Aug) or partial months of summer. For 1743, the degree of warmth/cooling and the trend of warmth/cooling in the series were investigated. (1) The degree of warmth/cooling in 1743 was expressed by a grade. We divided the reconstructed values of a series over the entire Little Ice Age into four grades by quartiles. The Little Ice Age was defined by the Encyclopedia Britannica as occurring from 1301-1850. If the reconstructed sequence was shorter than this period, the calculation was based

370    on the actual start and end times of the series. (2) The warm/cold trend was calculated by an 11-year (before and after 1743) trend.

    The following map displays the degrees of warmth/cooling and the trend of the 1743 summer event in different reconstructed sequences in the Northern Hemisphere (Fig. 6). The colour of the circles indicates warm summers in Northeast Asia and North America, while Europe was uniformly cold, which may have been influenced by extreme cold spring events in Europe in the 1740s (Brázdil et al., 2018; Brönnimann and Brugnara, 2023). The decadal trend in most places was warming,

375    which is consistent with the integration of cold and warm reconstructions in the Northern Hemisphere over the past millennium (Emile-Geay et al., 2017). The temperature increased in 1743, which was during the Little Ice Age. More climatic records from East Asia also support this phenomenon. For example, climatic records from both Japan and Beijing indicate that the 1740s were the beginning of the warm period (Aono and Kazui, 2008; Liu and Fang, 2017; Aono and Nishitani, 2022).

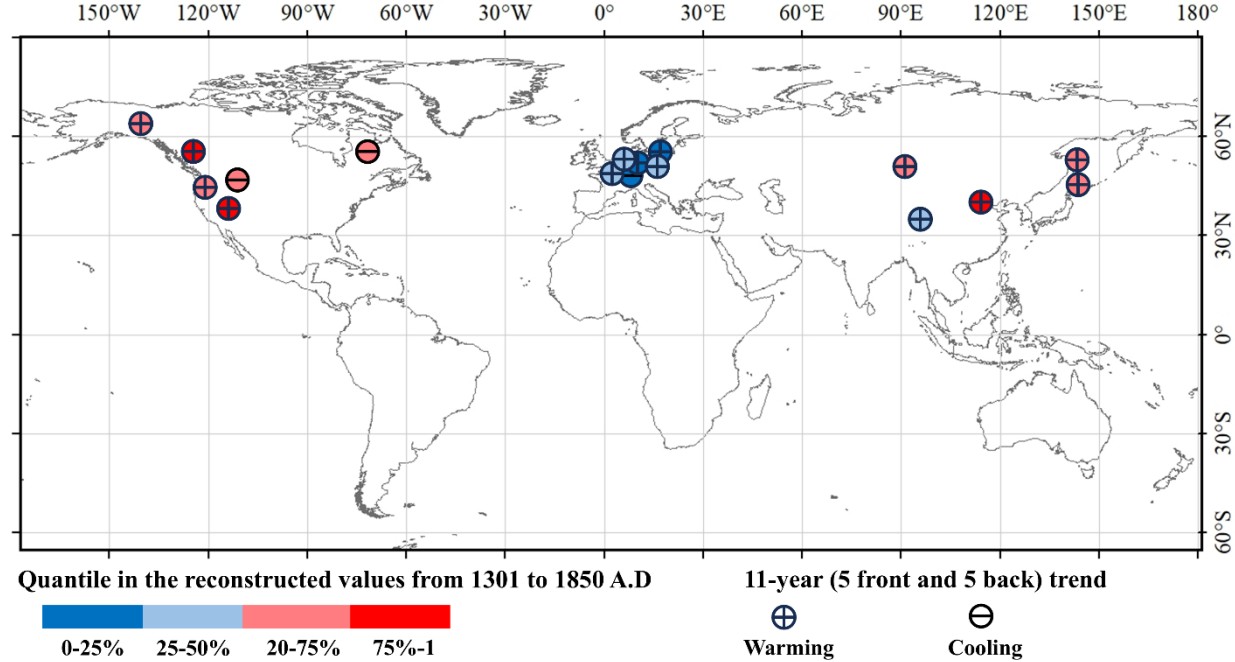

**Figure 6: Behaviour in the rest of the Northern Hemisphere during the summer of 1743.**

(Davi et al., 2002; Chuine et al., 2004; Meier et al., 2007; Glaser and Riemann, 2009; Dobrovolný et al., 2010; Wetter and Pfister, 2013; Wilson et al., 2014; Wang et al., 2015; Wiles et al., 2015; Luterbacher et al., 2016; Wilson et al., 2019; Martin et al., 2020; Davi et al., 2021; Heeter et al., 2021; Wang et al., 2022; Heeter et al., 2023).

However, it is difficult to determine whether short-term heatwaves occurred in Europe and North America due to the lack of simultaneous high-resolution textual or instrumental records in other regions. It is also unclear whether the fluctuations in the westerlies had a chain response in the middle and high latitudes of the Northern Hemisphere. Therefore, we believe that the 1743 event may have only been a very extreme heat event on a regional scale during a relatively warm period. This was unlike the large-scale extreme heat events and heatwaves in the Northern Hemisphere caused by global warming in recent years.

**5. Conclusion**

In this research, we utilized 3 kinds of historical materials, including local annals, archives and official history books, using methods of textual analysis, grading, and classification to investigate the temporal development, spatial pattern and characteristics of the impacts of extreme heat in 1743. We further inferred the range of extreme temperatures based on textual information and analysed the chain of impact transmission. The spatial characteristics of extreme heat and the number of deaths

caused were compared with those of typical modern examples. We also examined climate performance in other regions of the Northern Hemisphere in 1743.

The results showed that most of the Northern Hemisphere experienced an interdecadal temperature increase during the summer of 1743, with East Asia and North America on the warm side. The extreme heat of 1743 in North China occurred during this climate background.

1) The extreme heat of 1743 gradually increased. On June 22, people became aware of the unusual heat; after July 8, the heat continued to develop; beginning on July 14, human deaths caused by the heatwave began to be recorded; and beginning on July 19, the severe effects of the heatwave were given high priority by the central government. The heat peaked on July 25 and was later relieved by precipitation. The maximum daily temperature in 40 counties was likely to have reached 40-43 °C.

2) The extreme heat events of 1743 occurred on the on the plains of Hebei and Shandong Provinces, and in the valleys of southwestern Shanxi. The worst heat was distributed on the plains east of the Taihang Mountains, with the areas around Baoding, Shijiazhuang, and Xingtai being the main cities affected. They are also at high risk for heatwaves on the North China Plain in modern times.

3) The extreme heat event of 1743 caused damage to human health, plants and animals, and facilities. Of these, the impacts on crops persisted in human social systems. The number of deaths from extreme heatwaves has decreased compared to that in the historical period, but population deaths remain the most significant impact event of extreme heat events. Timely cooling and reducing exposure have been limited but necessary means to address high temperatures in both ancient and modern times.

**Data availability**

All original textual records in this study have been digitized and organized in Supplement 1.

**Author Contributions**

**Le Tao** and **Yun Su** contributed to the study conception and design. Funding acquisition was performed by **Yun Su**. All
historical documents are acquired, digitized and processed by **Le Tao**, and mapped by **Le Tao**, **Xudong Chen** and **Fangyu Tian**. **Xudong Chen** provided the writing outline and **Fangyu Tian** shared ideas to the discussion. The first draft of the manuscript was written by **Le Tao** and all authors commented on previous versions of the manuscript. All authors read and approved the final manuscript.

**Conflict of Interests**

The authors declare that they have no known competing financial or non-financial interests or personal relationships that could have appeared to influence the work reported in this paper.

**Acknowledgement**

We would like to thank all colleagues in our research group who provided valuable discussions and suggestions. We would like to thank the reviewers and editors for their valuable comments.

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
