# Peer review of "Processes, spatial patterns and impacts of the 1743 Extreme heat event"

_Climate of the Past, 2024_

## Referee Comment (RC2)

Process. Spatial pattern and impacts of 1793 extreme heat: from the perspective of historical documents.

Authors: Le Tao, Yun Su, Xudong Chen, Fangyu Tian

General comments

I liked this article for the meticulous search of information to characterize the severity and the regional extend of an intense heat wave that occurred in northern China almost 300 years in the past. I suggest to put in the title the location of the episode (northern China).

In my opinion the text needs some editing by someone fluent in English.  I am not, but I realized that in some paragraphs the language is not correct or it is difficult to follow.

Lines 36 – 37: the phrase "there are few records" seems unfinished.  Few records of what ?

Lines 75-76: "The annual average daily maximum temperature… in July is…"  Eliminate annual.

Line 86: "(Zhang 2004)". It should be "Zhang and Demaree 2004)

Line 106: Xiao et al 2012 is not in the References

Line 121: Explain what is Yu-Fen-Cun

Line 200: Explain what is Yu-Xue-Fen-Cun

Line 276: Fang et al. 2015 ia nor in the References

Line 328: Gasparrini et al. 2015 is not in the References

Line 329: Cheng et al. 2019 is not in the References

Line 359: The four citations in this line are not in the References

Line 359 – 360: "Because they often occur at the same time as westerlies fluctuate more." This phrase needs a better explanation

Line 362 – 363: "It is also unclear whether the fluctuations of the westerlies…"  N-S fluctuations ?  fluctuations in intensity?  This whole paragraph (lines 358 – 366) needs a better explanation.  What is the link between the westerlies and the heat waves in China ?

Lines 370 - 371: "in mining the climate eigenvalues"  I don't understand the meaning of this.

Table 5. 2003 France  15 days: 8.4 – 8.15 . These are 12 days, not 15 days.

References

Some of articles of the same principal author are not sorted by the year of publication (Brazdil et al.; Ge et al; Heeter et al, Wang et al, Wilson et al.

In the reference of Udías 1994, the name of the journal is missing.

---

## Author Comment (AC1)

[Figure]

**Figure 1: Study area and data distribution in this study**

(a) Location of the study area in China; (b) Study area and data distribution; (c) Temporal scenarios of

the number of hot records and drought and flood index in the study area, 1500-1911(Tao et al.,

2021;Hao et al., 2022).

---

## Author Response (AR1)

**Response to the comments on "Process, spatial pattern and impacts of 1743 Extreme heat: From the perspective of historical documents"**

Dear Editors and Reviewers,

Thank you very much for taking your time to review this manuscript. We really appreciate all your comments and suggestions. They have enabled us to greatly improve our work. According to the comments, we did some modifications. Please find our itemized responses below. The revised manuscript is attached, and the main modifications were marked in colors.
Thanks again!

*Comments from referee#1:*

We are very grateful to reviewer for reviewing our manuscript and giving valuable suggestions. Your suggestions will play an important role in improving the quality of our manuscript and inspire our future research.
We responded the comments with explaining our revisions point by point as follow.

*1. It is necessary to state what has been done by previous researches, e.g. Zhang & Demaree (2004), and the academic significance of this study.*

We have clarified our goal and emphasized the difference between our study and Zhang's by reorganizing the sentences in section 1:

1) We have **adjusted the narrative logic of the introduction**, to highlight the need for extreme heat reconstruction using historical documentary records. This can be reflected by the first sentence of each paragraph. (pp 1-2, Line 22-23/31-32/44-45/55-56, highlighted by yellow)

2) We have **furtherly analyzed the limitations and research gap** of Zhang's study. This can be seen in the third paragraph. (pp 2, Line 45-52, highlighted in red)

3) We **highlighted the importance of textual analysis methods** and their usability testing in historical extreme heat research in the fourth paragraph. **This is where our research differs from Zhang's in its focus** (pp 2-3, Line 55-56/ 70-71/ 73, highlighted by yellow)

The research of Zhang & Demaree (2004), including its preciousness and inspiration for our study, have been presented and analyzed in the introduction. As you mentioned, this study was critical and important, but its data were sporadic and early instrumental data. Their study was fortunate and fortuitous. The main advance of our study is to explore some feasible ways of analyzing the textual records of extreme heat events, while obtaining more information that have received less attention in previous studies from the records.

*2. Crop failures were more possibly caused by droughts and hot wind, rather than extreme heat.*

Our main revisions in response to this comment are as follows:

1) In section 2.3.3 we **emphasized the selection criteria of agricultural impacts records**, which means that only if the text clearly stated that the crop was damaged by heat was it counted as category (2). Also, we **added an explanation of the mechanism** of crop failure causation. (pp8, line 185-189, highlighted in yellow)

2) In section 4.2.1 we also emphasized the nature of heat-caused disasters in the chain of impacts, which is crop damage caused by concurrency with droughts or dry and hot winds. (pp 14, Line 283-28, highlighted by yellow)

We thank the reviewer for this comment, which have deepened our understanding of the agricultural disaster mechanisms of heat and drought.

Since this study starts from the textual records, we have categorized and discussed the various impacts that appear in the text. Crop failure is a specific type of impact that can be identified separately from text and is therefore presented and further discussed. In other words, we were faithful to the textual record for the extraction of impacts, but the extent of impacts was difficult to quantify (even in current botanical research). Human life and health impact are still the most dominant and serious, and are the focus of our attention.

*3.    Is it possible to compare the heat wave in northern China with the historical records of other East Asian countries? It would be helpful to better understand the weather background of this event.*

We thank the reviewer for this comment, which help us to better achieve the goals of the discussion in 4.3.

We **found some phenology evidence from Japan and China** and have supplemented them in section 4.3. They corroborated improved that 1743 was at the beginning of a warm period on long time scales, especially in east Asia. (pp 17, line 371-373, highlighted by yellow)

*4.    In order to give a temporal scenario of historical heat wave of the study area, it is necessary to establish the chronologies of heat wave and precipitation, though it is understandable that the historical records about heat wave are not complete. A comparison of these two chronologies would be helpful to better understand the frequency and climate background of historical heat wave.*

We have **added two time series about study area in Figure 1(c)**, they are the series of number of extreme heat records, and, the drought and flood index (both are from 1500 to 1911 CE in north China). **A description** of the climate change background of the study area and the particularity of year 1743 **was also added to the text** (pp 3, line 81-85, highlighted in yellow).

Thank you very much for your suggestion, which will be of great help to readers in understanding the context of climate change in North China during the historical period and in understanding the specificity of the case.

**Comments from referee#2:**

*General comments*

*I liked this article for the meticulous search of information to characterize the severity and the regional extend of an intense heat wave that occurred in northern China almost 300 years in the past. I suggest to put in the title the location of the episode (northern China).*

We are very grateful to reviewer for careful reviewing and valuable suggestions. Your suggestions will play an important role in improving the quality of our manuscript and inspire our future research.

We responded the comments with explaining our revisions point by point as follow.

*In my opinion the text needs some editing by someone fluent in English. I am not, but I realized that in some paragraphs the language is not correct or it is difficult to follow.*

We have **added the LOCATION** (in North China) in the title. (pp 1, line 2, highlighted in rose red)

We also accept your comments on the English writing, and we have **had the language touched up and polished** by a specialized institution after all the technical revisions are completed.

Thank you very much for your comments and suggestions again, your generous advice will be of great help to us in improving our paper.

*Lines 36 – 37: the phrase "there are few records" seems unfinished. Few records of what?*

Thank you for the reminder that this sentence fails to convey the full semantics. We originally meant to show that records of heat waves are rare compared to droughts and cold winters.

**We have revised the entire sentence** to "Compared with 227 extreme droughts in North China during 137 BC-2000 and 76 extreme cold winters in South China during 1500-1950, there are few records of heatwaves." (pp 2, line 37-38, highlighted in rose red)

*Lines 75-76: "The annual average daily maximum temperature… in July is…" Eliminate annual.*

Thank you for pointing out the repetition in the sentence, we have **removed the word ANNUAL** in the new manuscript.

*Line 86: "(Zhang 2004)". It should be "Zhang and Demaree 2004)*

Thank you for your careful reviewing. Please allow me to explain properly. This is another reference, titled "The General Collection of Chinese Meteorological Records for the Third Millennium", which you can see in the list of documents at the end of the article (fourth from the end). This book is also the source of data for our study, so there is no problem with the citation here.

*Line 106: Xiao et al 2012 is not in the References*

We are appreciated of your review and reminders.

It has been **added to the list of references**.

*Line 121: Explain what is Yu-Fen-Cun*
*Line 200: Explain what is Yu-Xue-Fen-Cun*

Thank you for your suggestions.

We have **added their explanation in a footnote**: Yu-Xue-Fen-Cun is a kind of agricultural weather record from Qing dynasty archives, which recorded the depth of infiltration after each precipitation (Yu-Fen-Cun) and the snow depth (Xue-Fen-Cun). It can be obtained from the First Historical Archives of China. The system description and applications of Yu-Xue-Fen-Cun can also be found in Ge et al, 2005. (pp6, in footnotes)

*Line 276: Fang et al. 2015 ia nor in the References*
*Line 328: Gasparrini et al. 2015 is not in the References*
*Line 329: Cheng et al. 2019 is not in the References*
*Line 359: The four citations in this line are not in the References*

I appreciate your careful review and reminder. It is possible that the manuscript was in the process of revision and the literature management software did not perform a good link migration. I regret and apologize for my oversight and for the review obstacle it caused.

After verification, **we have added the above missing sources (8 in total) to the literature list**.

*Line 359 – 360: "Because they often occur at the same time as westerlies fluctuate more." This phrase needs a better explanation.*

Thank you for your review and suggestions, which can well help us to enhance the logic and readability of the paper. At the same time, we also think that some of **the textual expressions in 4.3 do not express the intention of our discussion well**.

After consideration, we have **deleted the original statements** L358-L361(original preprint), and delete the first two sentences L335-L337(original preprint) of 4.3. We **added a new paragraph**. (pp16, line 346-355, highlighted in rose red)

We hope that such modifications would better introduce and clarify the purpose of this part of the discussion.

*Line 362 – 363: "It is also unclear whether the fluctuations of the westerlies…" N-S fluctuations? fluctuations in intensity? This whole paragraph (lines 358 – 366) needs a better explanation. What is the link between the westerlies and the heat waves in China?*

Thank you very much for your comment. We think this issue will also be better resolved with the modification above. The link between the westerlies and the heat waves in China was **explained in pp16, line 349-351**.

*Lines 370 - 371: "in mining the climate eigenvalues" I don't understand the meaning of this.*

The original intends of the phrases you mentioned is: using the description of the text to do some further information mining, as done in 4.1.1, to infer some meteorological feature values. At the same time, we found the advantages and disadvantages of textual records.

However, we realized that discussing the strengths and weaknesses of the textual record was not the key. Also, our English also hinders the expression of the meaning. We have **revised this sentence to:** "We further inferred the range of extreme temperatures based on textual information and analyzed the chain of transmission of impacts." (pp18, line 389-390, highlighted in rose red) This seems to **summarize what we've done more directly and ties in better with the conclusion**.

Thank you very much for your reminder, which is very helpful for us to improve our expression and summarize our research content.

*Table 5. 2003 France 15 days: 8.4 – 8.15. These are 12 days, not 15 days.*

Thank you very much for your careful review and kind reminder.

We have **checked the source paper** and found that the date range of 8.4-8.18 was wrongly noted as 8.15. The total of 15 days is correct. We have **corrected this**.

*References*
*Some of articles of the same principal author are not sorted by the year of publication (Brazdil et al.; Ge et al; Heeter et al, Wang et al, Wilson et al.*
*In the reference of Udías 1994, the name of the journal is missing.*

Thank you again for your careful review and generous reminder.

For the sort of the list, we have **reorganized it by year**.

We have **added the missing name of the journal in the reference** of Udías 1994. (pp23, line 572, highlighted in rose red)

We have also **rechecked** the references section and **corrected errors** in accordance with the requirements of the journal.

***Comments from community:***

We are very grateful to the comments and suggestions. Your suggestions will play an important role in improving the quality of our manuscript and inspire our future research.

We responded the comments with explaining our revisions point by point as follow.

*Major revisions:*

*1. Section 2.2.1- The authors refer to a total of three types of data sources. Was this hot event not reported in other records? The reasons for choosing these three categories need to be explained. In addition, it appears from the results that Data 2 and Data 3 were only used to illustrate the progression of the heat. Data 3 provided only 2 records, which is a very small number and perhaps needs to be accounted for.*

Thank you very much for your comment, which can help better explaining the data of the study.

We referred to the more commonly referenced documents or compilation on climate reconstruction in China during the Qing Dynasty. These three sources cover both local and central perspectives.

R3 were staged submitted, so the records are less compared to the day-by-day record of R2. However, both R2 and R3 have more precise temporal properties and so have been used primarily in reconstructing progression of the heat. (has been mentioned in pp5, L117)

Our modifications are as follows.

1) In the first paragraph of 2.2.1 of the paper we have **explained the choice of the three sources**. (pp4, line94, highlighted in bright blue)

2) We have **explained why R3 records are less**. (pp5, line122-123, highlighted in bright blue)

3) We **further described the characteristics and advantages of the three types** of information. (pp6, line126-129, highlighted in bright blue)

*2. Table 3 - How was the last column "Timescale of impact" obtained? This is not clearly described in the text. Correspondingly, how is Figure 4(e) plotted? This is a very important result. I think it is necessary for the authors to explain how it was converted from a textual record to a graph.*

Thank you very much for your comment, which are helpful in refining our methodology.

We have **added clarification** in section 2.3.3. (pp8, line188-192, highlighted in bright blue)

*3. 4.1.1 The authors give a demonstration of inferring temperatures from the coolness index. This is an interesting attempt. However, it is rather brief in discussing the limitations of textual records when it comes to physical quantification. A point to note is perhaps that body temperature is a composite perception of the environment and is influenced by factors other than temperature. There are also individual differences in body temperature between different people. This is not only limitation of the textual record.*

Thank you very much for your suggestion, which helps to improve the rigor of our discussion.

We have **provided additional clarification** in our discussions. (pp13, line274-276, highlighted in

bright blue)

*Minor Revision:*
*4. P14 . L298 - The meaning of "Tmax" needs to be explained.*

Thanks for the reminder, **we have explained it in the footnote**. (pp15, in footnote, highlighted in bright blue): Tmax refers to the maximum value of the daily maximum temperature in the month.

*5. Table 5 - Abbreviations for the serial numbers of the headings is used, which is inconsistent with the other tables, so please note to revise.*

Thank you for pointing out the errors in the manuscript, we have **corrected "Tab.5" to "Table.5"**.

---

## Author Response (AR2)

**Response to the comments on "Processes, spatial patterns and impacts of the 1743 Extreme heat event in North China from the perspective of historical documents"**

We really appreciate the reviewer of all your comments and suggestions. They have enabled us to greatly improve our work. We also thank the Editor very much for taking time to track and review this manuscript. The revised manuscript is attached, and the main modifications were marked in colors.

Thanks again!

*The authors have carefully revised the manuscript following the reviewers' comments, and significantly improved the readability of the manuscript. I therefore don't have further concerns. However, there are a few minor issues need to be modified.*

We are really very grateful to reviewer for careful reviewing again and giving new valuable suggestions. These comments help making our manuscript more rigorous.

In response to these suggestions, our modifications and clarifications are as follows:

*1. Several areas and place names are mentioned in the abstracts and texts, but are not shown in Figure 1. For example, where are 'the valleys of southwestern Shanxi', 'the plains east of the Taihang Mountains∏ and 'North China Plain'? please clarify and add them in Figure 1.*

1) They **have been clarified in Figure 1**: 'Fenhe Valley', 'North China Plain' and 'Taihang Mountains' **were labeled in Figure**. (pp 4, line 90).
2) The **text in 2.1 has also been modified** to more clearly introduce the geographical conditions of study area. (pp3, lines 79-82, highlighted in yellow)

*2. The yellow river in 1743 was significantly different from that of modern times, as the channel of the Yellow River migrated significantly in 1855. It is necessary to add the old channel before 1855 in figures 1 and 2. In addition, please avoid use the term 'plains north of the Yellow River', in order to avoid misunderstanding.*

1) The old channel before 1855 of the Yellow River has been added in Figure 1. (also in Figure 3, and 4).
2) Text has been added to 2.1 to simply describe the old channel of the Yellow River. (pp 3, lines 83-84, highlighted in yellow)
2) The term 'plains north of the Yellow River' **has been substituted in** 'the plains of Hebei and Shandong Provinces', **describing with administrative region**. (pp1, line 15, highlighted in yellow)

*3. Figure 4 (a~e), the legends are too brief and not clear. Please either modify them or clarify the meanings of the legends in the figure title.*

We have **modified the note under title** of this figure as 'classified by impact object (see Tab 3): (a) Human; (b) animal; (c) plant; (d) facility; and (e) time series of impacts', so that readers can understand what the legends mean according to table 3. (see pp 12, line 246, highlighted in yellow)

*4. Lines 329-330. 'During the Qing Dynasty, people in North China mainly engaged in agriculture, and long-term exposure to high temperatures was the main cause of death.' Was agriculture related to the deaths? Please rewrite this sentence.*

**The sentence has been rewritten** as 'During the Qing Dynasty, people in North China mainly engaged in agriculture, which means long hours of outdoor work even in summer. Long-term exposure to high temperatures was the main cause of death. Even indoors, the threat of high temperatures is high due to lack of cooling facilities.' (pp 16, lines 332-334, highlighted in yellow). We hope it will be clearer and more convincing.

*5. Figure 6. It is not necessary to include so many national boundaries, as they are usually politically controversial.*

The figure has been **replaced** by a new one **without national boundaries.** (see pp17, Figure 6)

*6. Please note the period of the Qing Dynasty at its first appearance.*

The period of the Qing Dynasty (1644-1911A.D) **has been noted at its first appearance**. (see pp2, line 44, highlighted in yellow)